# Feline Uveal Melanoma Review: Our Current Understanding and Recent Research Advances

**DOI:** 10.3390/vetsci9020046

**Published:** 2022-01-26

**Authors:** David Kayes, Benjamin Blacklock

**Affiliations:** The Ophthalmology Department, The University of Edinburgh Hospital for Small Animals (HfSA), Royal (Dick) School of Veterinary Studies (R(D)SVS), Easter Bush Campus, Midlothian EH25 9RG, UK; ben.blacklock@ed.ac.uk

**Keywords:** feline, uveal, melanoma, iris, melanosis

## Abstract

Melanocytic neoplasia is the most common form of ocular tumour in cats, accounting for 67% of cases in an analysis of 2614 cases of primary ocular neoplasia. Feline diffuse iris melanoma (FDIM) is by far the most common form of ocular melanocytic neoplasia, with limbal melanomas and atypical melanoma (melanoma affecting the choroid or ciliary body) infrequently recognised. Early lesions begin as flat areas of pigmentation of the iris, known as iris melanosis. This melanosis is a precursor lesion that can become FDIM when pigmented cells infiltrate the anterior iris stroma, commonly alongside a transition in cell morphology. The differentiation between FDIM and benign iris melanosis is only recognisable though histologic examination, with no in vivo means of identifying the malignant transformation. The behaviour of FDIM is variable and difficult to predict. Some FDIM lesions have a more benign progression and can slowly grow or remain static for years without affecting the ocular or systemic health of the individual, whilst other tumours behave aggressively, invading the ocular structures and significantly affecting the life expectancy of cats through metastatic disease. This makes management and timely enucleation of these cases challenging in practice. This article aims to review our current knowledge of FDIM.

## 1. Introduction

The average age of cats affected with feline diffuse iris melanoma (FDIM) is 9.4 years [1] with a reported range from 2 to 23.1 years [2,3,4,5,6]. FDIM begins as hyperpigmented foci consisting of a proliferation of dysplastic melanocytes, which appear as flat, brown spots on the iris surface. Dysplastic features of the melanocytes were defined by Featherstone et al. (2019) as plump, pigmented melanocytes with varying degrees of anisokaryosis with or without hyperchromasia and/or discernible nucleoli [5]. These precursor lesions are considered benign and are known as iris melanosis, where the melanocytes are confined to the anterior iris in 1–3 layers, regardless of the degree of melanocyte atypia. The behaviour of these lesions is unpredictable. Some remain static or grow slowly over months to years, resulting in only a cosmetic change to the iris. Others may progress rapidly [2]. The transition of iris melanosis to early FDIM is only recognisable histologically, where invasion of the dysplastic melanocytes into the iris stroma occurs [7].

Amelanotic variants occur infrequently and may be mistaken clinically for inflammatory lesions [8,9]. FDIM progressively expands through the iris, iridocorneal drainage angle and ciliary body, eventually penetrating the sclera. Iris thickening, dyscoria and reduced pupil mobility and secondary glaucoma may occur secondary to tumour infiltration of the iridocorneal angle. The tumour may exfoliate into the anterior chamber, and dissemination of neoplastic cells via the aqueous humour is a suggested route of metastasis [9]. Extraocular metastasis is most likely through haematogenous spread via the scleral venous plexus. The sclera must be penetrated by neoplastic cells for lymphatic spread to occur, as the eye is devoid of lymphatics [8]. Metastatic disease most commonly occurs in the liver but also the lungs, kidneys, spleen, lymph nodes, brain and bone [1,2,3,4,8,10,11]. A latent period of several years between diagnosis of FDIM and death from metastatic disease has been reported [2,3,4]. There are wide variations of observed metastatic rates between studies, ranging from 19% to 63% [3,4]. Local recurrence in the orbit is also possible following enucleation [1].

Uveitis and lens capsule rupture have also been described as concurrent findings [2,4,8,12]. Enucleation is the standard treatment for FDIM, with no data on surgical iridectomy or photocoagulation using a neodymium:yttrium, aluminium, garnet (Nd:YAG) laser [10]. These treatments are not recommended in the treatment of FDIM cases due to the high metastatic potential and diffuse nature of the disease [13].

Experimental intracameral injection of feline sarcoma virus (FeSV) in kittens has been shown to induce the formation of metastatic melanoma of the iris and ciliary body [14]. The presence of naturally occurring feline leukaemia virus (FeLV) and feline sarcoma virus (FeSV) was later investigated by Stiles et al.; nested polymerase chain reaction (PCR) testing was performed on deparaffinised tissue sections from 36 formalin-fixed paraffin-embedded (FFPE) globes with FDIM. The study identified that 3 out of 36 globes were positive for FeLV-FeSV [15]. A further study by Cullen et al. failed to identify FeLV-FeSV on PCR and immunohistochemistry (IHC) testing of 10 FFPE globes with FDIM [16].

FDIM tends to remain limited to the anterior uvea, not expanding posteriorly. By contrast, feline atypical melanoma originates from any area of the uvea that is not the anterior iris. As the name suggests, it is infrequently recognised, representing only 1% of feline neoplasms in the comparative ocular pathology laboratory of Wisconsin (COPLOW) collection [1]. A characteristic feature of atypical melanoma is the aggregation of cells in the subretinal space [1]. Cases are often only recognised after advanced ocular infiltration occurs, due to the posterior origin of the neoplasm [17]. Histologically, cells have a uniform morphology and appear round and darkly pigmented with small round nucleoli. By comparison, advanced FDIM tumour cells are lightly pigmented and highly anaplastic. Atypical melanoma has a metastatic potential, despite a relatively bland cell morphology [1,17]. A single case of feline primary choroidal melanocytoma [18] and melanoma [19] have been reported. There is a single case report of a non-pigmented iris melanoma arising from a phthisical eye in a 13-year-old female domestic shorthair cat [20].

## 2. Diagnostic Evaluation

### 2.1. Introduction

Cats presenting with iris hyperpigmentation should undergo a full, routine ocular examination. Slit lamp biomicroscopy, gonioscopy and indirect ophthalmoscopy are essential to characterise and evaluate the extent of the lesions; as well as to identify any potential indicators of malignancy. As the dysplastic melanocytes first infiltrate the iris stroma and become early FDIM, lesions are initially indistinguishable from iris melanosis on ocular examination (Figure 1) [2,5]. However, as FDIM progresses, infiltration of the iris stroma leads to recognisable changes such as iris thickening, dyscoria, reduced pupil mobility, pigment dispersion in the anterior chamber and involvement of the iridocorneal angle [10].

Intraocular pressure measurement should be performed due to the risk of secondary glaucoma with tumour infiltration of the iridocorneal angle. FDIM is an important differential for cats presenting with unilateral glaucoma. Figure 2 is of a 12-year-old Domestic Shorthair that presented for evaluation of unilateral glaucoma. Subsequent enucleation with histopathology confirmed the cause to be FDIM.

Frequent re-examination and careful photographic documentation are recommended for slowly progressive lesions or modest changes [10].

Densely pigmented iridociliary cysts can be mistaken for melanocytic neoplasia. Fragola et al. reported on a case series of 14 globes that were mistakenly enucleated based on a suspicion of neoplasia: all 14 globes contained one or more iridociliary cysts and were free from neoplastic changes. Of these globes, eight were enucleated on suspicion of melanocytic neoplasia. The majority (*n =* 9) of globes were enucleated on the recommendation of board-certified veterinary ophthalmologists. The suspected neoplasia was thought to be within the iris in half of the cases (*n* = 7) [21].

Iridociliary cysts may appear at the pupil and originate from the posterior iris epithelium. In most cases they are an incidental finding, not causing a clinical problem [22]. Iridociliary cysts can result in dyscoria and may also occur in the contralateral eye.

Ocular ultrasonography can help differentiate between a melanocytic mass and thick-walled, densely pigmented iridociliary cysts. It can also aid characterisation of the extent of an ocular neoplastic mass [13].

Advanced cross-sectional imaging such as computed tomography (CT) or magnetic resonance imaging (MRI) are also recommended imaging modalities when there is a suspicion of FDIM, in order to ascertain the involvement of the periorbital bony structures, as well as screening for the presence of metastatic disease [13].

### 2.2. Iris Biopsy

As histological examination is required to differentiate early FDIM from iris melanosis, appropriate management of cats with hyperpigmented iris lesions can be a significant challenge for clinicians. Iris biopsy has been described to be a useful diagnostic technique to differentiate between an early FDIM and iris melanosis in order to justify early enucleation of cases of FDIM [5].

Featherstone et al. (2019) described the technique and complications in a retrospective review of seven cats that underwent iris biopsy under general anaesthesia and neuromuscular blockade. Between one and six biopsies were taken from each iris, with partial thickness samples in 4/7 cases and full thickness samples in the rest of the cases. Only minor complications were identified including mild, self-limiting iris haemorrhage (*n* = 4), the formation of a fibrin clot in the anterior chamber (*n =* 2), pseudopolycoria (*n* = 2), corneal ulceration (*n* = 1) and dyscoria (*n* = 1). Post-operative ocular hypertension occurred in both eyes of one cat that underwent concurrent phacoemulsification surgery. Two biopsies were consistent with FDIM and resulted in enucleation of the globe. Histopathology of the enucleated globes confirmed FDIM in both cases [5].

### 2.3. Histologic Evaluation

Morphologic characteristics are usually sufficient for diagnosis of feline uveal melanomas. Histologic indicators of melanocyte malignancy include an increase in mitotic figures, a higher nuclear: cytoplasmic ratio and an increase in nuclear pleomorphism (including mononuclear gigantism and multinucleation). However, no clear cut-off value exists for these criteria.

FDIM begins as a cluster of small, angular pigmented cells on the anterior iris surface. These precursor lesions are considered benign regardless of the number of dysplastic features displayed [5]. They are only considered malignant once the dysplastic melanocytes begin to invade the iris stroma. This may coincide with a cell morphology shift from small, angular cells to rounded cells with a round nucleus and prominent nucleolus [7]. Figure 3 is a series of histological sections demonstrating features of melanocyte dysplasia as well as invasion of the iris stroma with neoplastic melanocytes.

FDIM displays a wide variety in cellular morphology including round or polygonal cell, spindle cell, balloon cell, anaplastic variants and giant cell types [8]. There is no correlation between metastatic disease and cellular morphology or presence of balloon cells [7]. Melanin bleaching of heavily pigmented samples may be required to reveal cytologic detail in order to detect mitoses and nuclear atypia [23].

Intraocular melanomas can show epithelial, round cell and mesenchymal characteristics, making morphologic diagnosis of poorly pigmented variants challenging [1,13]. Figure 4 demonstrates the gross appearance of an amelanotic variant compared to the more common pigmented variant.

Poorly differentiated FDIM may be mistaken for pleomorphic lymphoma, histiocytic sarcoma or anaplastic metastatic carcinoma [23]. Immunostaining with S-100, Melan-A, tyrosinase-related protein-2 (TRP-2) human melanosome-specific antigen-1 (HMSA-1) and HMSA-5 may aid in the diagnosis of poorly pigmented feline ocular melanomas [12,13,23,24]. The use of PNL2 staining has also been described but is associated with the degree of pigmentation and its use in poorly pigmented ocular melanomas has yet to be investigated in cats [25]. However, all of the stains have a variable sensitivity and specificity, and, therefore, multiple markers should be used to confirm a diagnosis of melanocytic neoplasia [13]. Grahn et al. (2006) found a 12% discordance between morphologic diagnosis of feline intraocular tumours and diagnosis based on histochemical staining and IHC labelling. The authors developed an algorithm for the diagnosis of primary and metastatic intraocular neoplasms in cats. For a diagnosis of melanoma, samples had to stain positive for vimentin and negative for periodic acid-Schiff (PAS). Additionally, samples had to either be positive for S-100 and tyrosinase or positive for HMSA5 and negative for CD3 and LY5 [12].

## 3. Factors Associated with Metastatic Disease

### 3.1. Histologic and Morphologic Characteristics

Early reports on FDIM found a high metastatic rate. Patnaik and Mooney (1988) looked at 16 cats with a histopathological diagnosis of FDIM based on enucleation samples. Follow-up information was available for 14 cats. Metastatic disease was found in 63% of cats and confirmed on postmortem examination and histopathology. All 16 cases had advanced neoplastic infiltration of the globes. In six cases, the tumour involved the iris and ciliary body; in four cases, the tumour ‘extended to the sclera, cornea, chambers of the eye and beyond’, and in four cases, ‘the whole globe was replaced by the neoplasm’ [3]. Although a high metastatic rate was observed by the study, many of the globes clearly had advanced disease at the time of enucleation.

Several histological characteristics have since been reported to be associated with the rate of metastatic disease. Kalishman et al. (1998) showed that the degree of tumour extension was significantly associated with survival times. The study categorized cases into ‘early’, ‘moderate’ and ‘advanced’ groups. In ‘early’ cases (*n* = 9), tumours were confined to the iris and trabecular meshwork. Cases in this category had comparable survival time to control cats. ‘Moderate’ (*n* = 12) was defined as tumour in the iris, rostral ciliary body but not the sclera. ‘Advanced’ (*n* = 13) cases had tumour throughout the ciliary body and extending into the sclera. Increasing tumour grade was significantly associated with a decreased survival time after adjustment for age (*p* = 0.007). This provided some evidence to support early enucleation of cases with FDIM. However, the study lacked data on the cause of death in most of the cases. Metastatic disease was presumed to be the cause of death based on a history of weight loss or a palpable abdominal mass [2].

A more recent work by Wiggans et al. looked at 47 enucleated eyes with a diagnosis of FDIM and analysed the prognostic value of various histological and immunohistochemical characteristics. Histological analysis was followed by immunolabelling of the sections against melan-A, B-Raf, PNL2 and epithelial cadherin (E-cadherin) [4]. The study supported previous findings that local tumour invasion is associated with an increased rate of metastasis. Extrascleral extension (*p* = 0.039), choroidal invasion (*p* = 0.045) and necrosis within the neoplasm (*p* = 0.026) were found to be significant risk factors for metastasis [4]. The study also confirmed previous findings of an association between a high mitotic index with a worse prognosis [8], with a cut off value given as >7 mitoses per high power field associated with an increased rate of metastasis (*p* = 0.024). Contrasting with previous reports, the study did not show an association between scleral venous plexus penetration and metastatic rate. There was no association between cell morphology and the rate of metastasis. The majority of tumours had a mixed epithelioid-spindle cell morphology (*n* = 31) followed by epithelioid cells only (*n* = 16) and the presence of balloon cells (*n* = 12). Metastatic disease was suspected in 9/47 (19%) cats: in 7/9 cases based on masses identified on abdominal ultrasonography (*n* = 6) and thoracic radiographs (*n* = 1); two cases had metastatic disease confirmed histopathologically on postmortem examination [4].

### 3.2. Immunohistochemical Markers

Wiggans et al. found that increased metastatic rates were associated with increased E-cadherin (*p* = 0.036) and Melan-A (*p* = 0.023) label intensity. A decreased metastatic rate was associated with increased PNL2 label homogeneity (*p* = 0.008). The study also observed that B-Raf was consistently expressed by FDIM cells but not by normal melanocytes [4].

Melan-A is an antigen expressed by normal melanocytes and increased expression has been found to be associated with increased survival times in humans. Human studies have shown differing expression patterns between cutaneous and ocular melanomas. Uveal melanomas showed homogenous expression but 17% of cutaneous melanomas lacked expression [26]. Melan-A is recognised by T-lymphocytes [27] and is therefore a potential target for immunotherapy.

Melanocytes form adherent and regulatory junctions to neighbouring cells via E-cadherin transmembrane proteins [28]. Reduced expression is associated with an increased metastatic rate in human cutaneous melanoma as well as other neoplasia [29]. By contrast, upregulation in human and canine uveal melanocytic neoplasia has been associated with an increased risk of metastatic disease [30,31]. FDIM behaviour more closely mimics the behaviour of uveal melanomas than cutaneous melanomas in these species [4,30,31].

B-Raf is the product of the oncogene serine/threonine kinase (BRAF), with gain of function mutations identified in uveal melanomas of humans [4].

### 3.3. Circulating Cell-Free DNA (cfDNA)

cfDNA is non-encapsulated DNA that enters the bloodstream following apoptosis or necrosis of cells. cfDNA levels have been shown to be of diagnostic and prognostic value in humans and dogs with cancer and other pathologies [32].

Rushton et al. (2019) showed that cell-free DNA (cfDNA) was not of use as a diagnostic or prognostic tool for FDIM. There were no significant differences in cfDNA concentration or integrity index (*p* > 0.01) between cats with iris melanoma (*n* = 34), iris naevi (*n* = 30) or the control groups with no melanocytic abnormalities of the iris (*n* = 32) [33].

The study may have been limited due to the small amount of cfDNA extracted from the plasma of cats compared to humans due to the relatively small volume of plasma obtained. Histology was only performed on 24 globes that were enucleated following a suspicion of FDIM. The other globes were diagnosed based on clinical examination alone and hence the results may be prone to misinterpretation. The study suggested that the lack of a breakdown of the blood aqueous barrier, in all but extremely advanced cases of FDIM, may prevent significant levels of cfDNA reaching the bloodstream. In humans with retinoblastoma, aqueous humour liquid biopsy has been shown to yield superior amounts of tumour-derived cfDNA to that of blood [34]. cfDNA obtained from the aqueous humour was associated with therapeutic response of retinoblastoma. Tumour progression was associated with a higher tumour fraction (TFx) (the proportion of cfDNA that is derived from the tumour) and tumour-derived somatic copy number alterations (SCNA) (*p* ≤ 0.04) [35].

Future research into the presence of cfDNA in the aqueous humour of cats with ocular melanomas is suggested [33].

### 3.4. The Role of Tumour Infiltrating Lymphocytes (TILs)

TILs are host immune cells directed against neoplastic cells. Their presence has been associated with an improved prognosis in melanomas. Multiple studies have also shown their significance as therapeutic targets and response to cancer immunotherapy [36].

A preliminary report by Porcellato et al. investigated the association between TILs and immunohistochemical markers and histological features. Thirty-six cases of feline melanomas were selected [25]. Of those, 12 were ocular melanomas, 18 cutaneous melanomas (2 of which were melanocytomas) and 8 oral melanomas. The cases underwent histological evaluation before immunohistochemistry staining was performed with antibodies for S100, melan-A and PNL2 to confirm the histologic diagnosis. Cases had to be positive for S100 and either melan A or PNL2 to be confirmed as melanocytic neoplasms and included in the study. Evaluation of TIL presence, distribution and density was performed before samples underwent immunohistochemistry staining with CD3 and CD20 Rabbit polyclonal antibody markers. TILs were shown to be frequently found in the melanocytic neoplastic microenvironment of cats. Higher TIL grades were associated with lower expression of Melan-A and PNL2 (*p* < 0.05) as well as the percentage of necrosis in ocular melanocytic tumours [36]. The results suggest that TILs may be associated with histologic features of malignancy and loss of Melan-A and PNL2 expression.

### 3.5. Secondary Glaucoma

Kalishman et al. found that secondary glaucoma was a negative prognostic indicator for survival in cats with FDIM. However, the age-adjusted value for glaucoma and survival was not statistically significant (*p =* 0.07) [5]. A more recent study by Wiggans et al. did not find a statistically significant relationship between the presence of secondary glaucoma and metastatic disease (*p* = 0.36) [7].

## 4. Gene Expression and Mutational Analysis

Rushton et al. (2017) performed a pilot study into the genetic landscape of ocular melanomas by investigating possible gene mutations as well as the gene expression status of the genes most commonly involved in initiation and progression of uveal melanomas in humans and dogs. DNA sequencing (*n = 10)* and RNA expression levels (*n* = 10) were measured through quantitative real-time polymerase chain reaction (RT-qPCR) from 12 samples of ocular melanoma (11 iris melanomas and 1 conjunctival melanoma) [37].

DNA sequencing was performed with polymerase chain reaction (PCR) primers for the most commonly identified mutations identified in human cutaneous and uveal melanomas (*BRAF^V600E^*, *GNAQ^Q209^*, *GNAQ^R183^*, *GNA11^Q209^*, *NRAS^G12/13^*, *NRAS^Q61^*, *KIT^W557^*, *KIT^K642^*, *KIT^D816^* and *MEK1^I111^*). These mutations were not identified in feline melanoma samples [37].

RNA expression analysis was performed on *GNAQ*, *GNA11*, *BRAF*, *KIT*, *NRAS*, *BAP1*, *FXR1*, *LTA4H*, *RASSF1* and *CDH1* genes. A significant (*p* < 0.05) upregulation of *KIT* and *LTA4H* and a downregulation of *GNAQ*, *GNA11*, *BRAF* and *RASSF1* was identified [37].

Rushton et al. developed their previous work by measuring the protein expression of the previously studied genetic mutation hotspot regions. The expression of *KIT*, *BRAF*, *GNA11*, *GNAQ* and *RASSF1* was analysed in 57 formalin-fixed paraffin-embedded (FFPE) globes of histologically confirmed iris melanomas and a control group of 25 FFPE globes that were free from ocular abnormalities. FFPE sections were stained with antibodies specific to the proteins under investigation using immunofluorescence. The FDIM globes showed an increased expression of *KIT*, *BRAF*, *GNAQ* and *GNA11* (*p* < 0.05) [38].

*BRAF* is a proto-oncogene (encoding for the protein B-Raf) that activates the mitogen-activated protein kinases/extracellular signal-regulated kinases pathway (MAPK/ERK) [39]. Mutations in BRAF have been reported to occur in 66% of human melanomas [40]. *GNAQ* and *GNA11* have various roles in the normal cell cycle, DNA synthesis and tumour extravasation in human and mice studies [29,30,31]. *KIT* encodes for receptor tyrosine kinase proteins, which play a key role in the cell signal transduction and the function and development of many cell types including melanocytes [41,42,43]. *KIT* overexpression has been demonstrated in human uveal melanomas [44].

The study demonstrated that FDIM shows increased expression of proteins that have been demonstrated to play a key role in the tumour progression pathway in humans (*KIT*, *BRAF*, *GNAQ* and *GNA11*). In doing so, the study has potentially identified multiple possible chemotherapeutic targets. B-Raf inhibitors are being investigated as adjunctive chemotherapy agents [7]. The most significant result of the study may be the identification of *KIT* overexpression in FDIM samples. Human melanoma xenografts and cell lines with *KIT* overexpression have been shown to be highly sensitive to the chemotherapeutic drug imatinib (a tyrosine kinase inhibitor) [45]. The identification *KIT* overexpression in FDIM may prove to be of fundamental importance in the development of adjuvant therapies for advanced FDIM cases.

However, the increased *BRAF*, *GNAQ* and *GNA11* protein expression shown by immunofluorescence does not support the previous pilot study by Rushton et al. (2017) which showed decreased mRNA expression of these genes [37,38]. The author hypothesises that complex post-transcriptional mechanisms may be involved in the differences seen; or that potential errors may have occurred in detection or evaluation of mRNA or protein using qPCR and IHC [38]. Further research is required to validate the findings of this study and to understand why differing mRNA and protein expression exist for *BRAF*, *GNAQ* and *GNA11*.

## 5. Adjuvant Therapy

To date, there are no recognised adjuvant therapies for metastatic disease of FDIM [37]. A xenogenic human tyrosinase DNA vaccine (Oncept^®^) was developed for use in canine oral menaloma [46]. The vaccine has been shown to improve the expected survival time of dogs with malignant melanoma [47,48]. The canine melanoma vaccine (Oncept^®^) has been shown to be safe to administer to cats, with minimal risk of side effects [49]. However, there are no data on the effect of vaccination on the expected survival times of cats with melanocytic neoplasia. Further investigation into the use of the vaccine as an adjuvant therapy in cats is required.

## 6. Discussion

The data on metastatic rates are relatively sparse and limited in several ways. Cases included in early research by Patnaik, A.K. and Mooney were extremely advanced at the time of enucleation, biasing towards findings of a high metastatic rate [6]. Case numbers were also severely limited. In later studies, metastatic disease was often not confirmed to be secondary to FDIM based on histopathological or cytological analysis [5,7]. Metastatic disease was assumed, based on clinical history or evidence of a mass in abdominal or thoracic imaging. These studies may falsely overestimate the true metastatic rate of the disease. Data provided by Kalishman et al. (1998) indicating that early enucleation resulted in an improved outcome, provides justification for clinicians to perform enucleation of visual and non-painful eyes [5]. This information is elucidated from the data of only nine cats with ‘early’ FDIM. Enucleation of a visual and non-painful eye is a challenging decision for any pet owner. It is important that accurate information is obtained on the risks of metastatic disease and the likely impact on survival of affected cases. Only once accurate information is obtained can owners of affected cats make a truly informed decision on whether to opt for an enucleation.

Iris biopsy provides the clinician with a useful diagnostic tool that may help identify cases of early FDIM. Results of iris melanosis from an iris biopsy sample must be received with caution for two reasons. Firstly, the sample may not be representative of all the affected tissue. Secondly, iris melanosis may later transform to FDIM and repeat biopsy may be indicated if iris lesions continue to progress. Validated immunohistochemical markers of prognosis, such as those described by Wiggans et al. (2016) could be applied to iris biopsies and potentially iris aspirates and aqueocentesis samples [7,9]. This would greatly increase the diagnostic yield of an iris biopsy.

Featherstone et al. (2019) also discusses the confusing nomenclature surrounding iris melanosis. Currently, the term ‘iris melanosis’ only accounts for the location of the melanocytic cells in the anterior iris stroma, regardless of the degree of atypia. It is likely that melanocytic cells with a high degree of cellular atypia are more likely to progress and infiltrate the iris stroma. The publication suggests adjusting the nomenclature in order to more accurately define these dysplastic melanocytes. This may be beneficial in managing owner expectations, especially in their awareness of cases that show a high degree of atypia and may be more likely to progress to FDIM [9].

Rushton et al. (2017; 2019) identified several changes in mRNA and protein expression of genes in FDIM that are known to play a key role in the initiation and progression of human and canine uveal melanoma [37,38]. However, data surrounding the genetic and transcriptomic data of FDIM are sparse, and further work is required to understand the genetic mechanisms responsible for the biological behaviour of FDIM involved in initiation, progression and metastasis. Transcriptomic and genetic data help predict the efficacy of a novel therapy by identifying molecular chemotherapeutic targets. Data provided may also improve the diagnostic yield outcome of surgical biopsy of iris tissue, improving prognostication of clinical outcomes and potentially avoiding early enucleation of cats affected with early FDIM or iris melanosis.

## 7. Conclusions

Cats presenting for evaluation of hyperpigmented iris lesions should undergo a full physical and ocular examination including gonioscopy. Frequent re-examination and careful photographic documentation are recommended for slowly progressive lesions or modest changes. Iris biopsy should be considered to help confirm a diagnosis of early FDIM but must be interpreted with caution if a result of iris melanosis is obtained. Ocular ultrasound and advanced cross-sectional imaging such as CT or MRI should be considered in cases of FDIM to evaluate the extent of tumour invasion and to ascertain involvement of the periorbital bony structures, as well as screening for the presence of metastatic disease. Enucleation is currently the only recommended treatment option for confirmed cases of FDIM. The difficulty in predicting disease progression and the limitations of the data surrounding the metastatic potential of FDIM should be discussed with pet owners. There are no data surrounding the effect of adjuvant therapies such as the melanoma vaccine on the survival time of cases with confirmed metastatic disease.

The data already provided on the metastatic potential of FDIM must be verified by prospective research that includes follow up data, metastatic staging and confirmation of metastatic disease based on cytology/histopathology of metastatic lesions.

Understanding the mechanisms that are responsible for the progression of iris melanosis to FDIM, and from early FDIM to advanced disease with a high metastatic potential, will prove invaluable to clinicians for accurate prognostication and identification of adjuvant chemotherapeutic therapies. The genetic and transcriptomic landscape of FDIM is an exciting opportunity for future research in this field.

## Figures and Tables

**Figure 1 vetsci-09-00046-f001:**
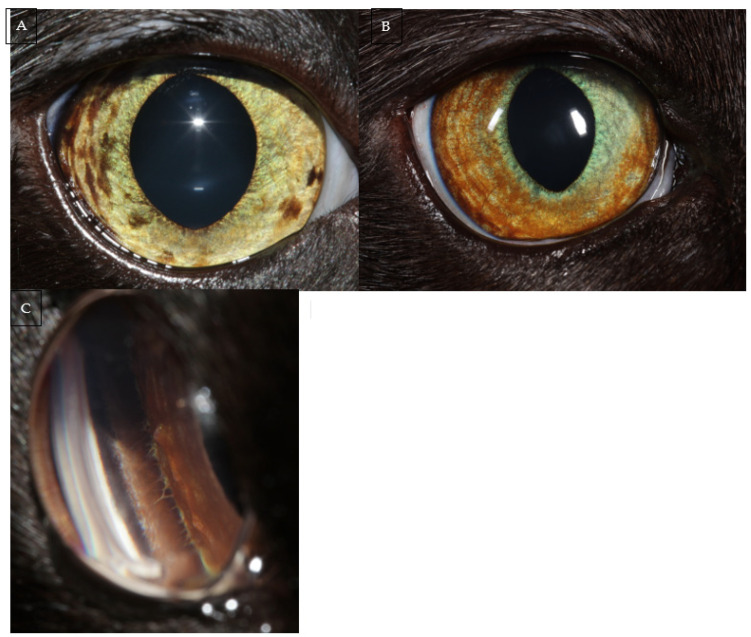
(**A**) Multifocal iris hyperpigmentation in a 4-year-old Domestic Shorthair. (**B**) Extensive, diffuse hyperpigmentation of the iris of a 7-year-old Domestic Shorthair with appearance of the iridocorneal drainage angle in the same eye (**C**). Gonioscopy is performed routinely for all cases presenting with a suspicion of feline diffuse iris melanoma (FDIM). The hyperpigmentation in both cases may represent iris melanosis or FDIM and iris biopsy should be considered.

**Figure 2 vetsci-09-00046-f002:**
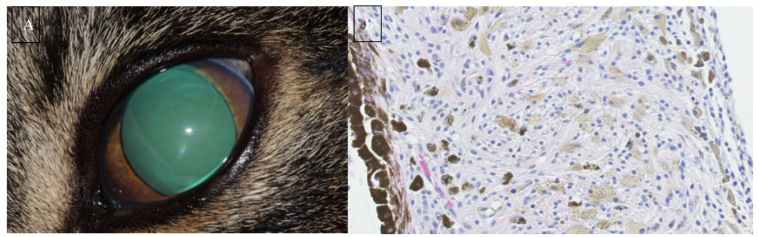
(**A**) Photograph of a 12-year-old Domestic Shorthair presenting for unilateral glaucoma. Subsequent enucleation resulted in a histopathologic diagnosis of feline diffuse iris melanoma. (**B**) A high magnification (400×) H&E section from the same case. Note the heavily pigmented posterior iris epithelium aiding orientation of the image. The iris is infiltrated with spindle to round cells. Most cells have a lightly pigmented brown granular cytoplasm with scattered highly pigmented cells. Nuclei are round to ovoid with coarsely stippled chromatin and often a single prominent nucleolus. Anisocytosis and anisokaryosis are mild and mitoses are rare which is not typical of FDIM. Image (**B**) courtesy of *Linda Morrison* and *Alexandra Malbon* (University of Edinburgh Pathology Department).

**Figure 3 vetsci-09-00046-f003:**
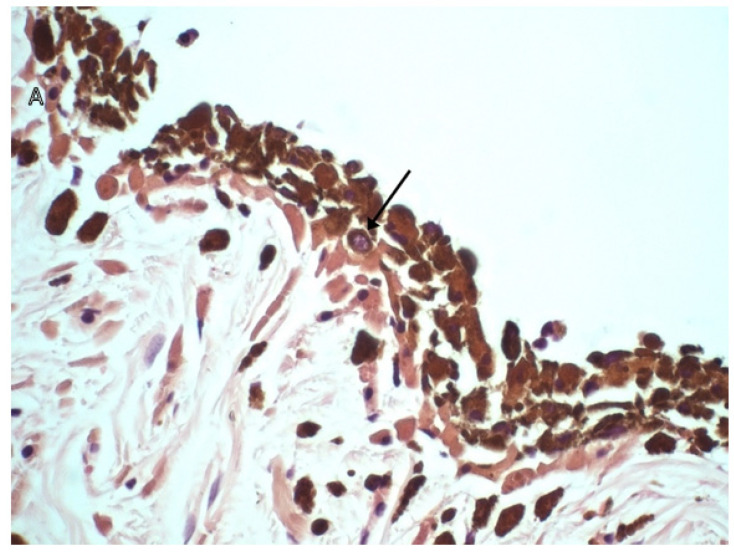
H&E histopathology sections of the feline iris. (**A**) A high magnification (×600) section showing a melanocyte with a prominent, open nucleus with a small distinct nucleolus, indicating nuclear atypia. (**B**) A high magnification (×600) section of an early case of FDIM with the very start of invasion of the stroma with dysplastic melanocytes. A low magnification (×100) (**C**) and a high magnification (×400) (**D**) section of two separate cases of FDIM. Dysplastic melanocytes with anisokariosis and nuclear atypia line the anterior iris stroma (arrows) and invade the iris stroma (circles). Images courtesy of *Emma J. Scurrell* (CytoPath Veterinary Pathology).

**Figure 4 vetsci-09-00046-f004:**
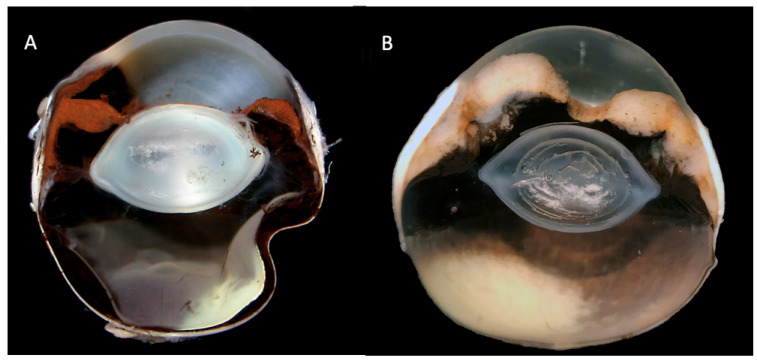
Gross section of a feline globe with diffuse iris melanoma (FDIM) (**A**) and an amelanotic FDIM variant (**B**). Note the marked iris thickening in both sections due to infiltration with neoplastic melanocytes. Images courtesy of *Emma J. Scurrell* (Cytopath Veterinary Pathology).

## Data Availability

The study did not report any new data.

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
