# Peer review of "Feline Uveal Melanoma Review: Our Current Understanding and Recent Research Advances"

_vetsci, 2022, doi:10.3390/vetsci9020046_

Round 1

Reviewer 1 Report

Thank you for this useful review in the state of play in FDIM. I have some minor comments but in the whole this is a good summary of the present knowledge.

Line 8- I would add "in cats" after ocular tumour.

Line 28 - Dysplastic rather than Dysplasic.

Line 76 - "lesions are initially indistinguishable"

Line 97 - "The majority" rather than just Majority

Line 104- "ultrasonography can also aid"

Line 148- there is a superscript 2. Is this reference 2?

Line 154 - Patnaik spelling error

Line 204- This sentence does not make sense. Antibody to an as yet unidentified antigen?

line 256 - where is figure 2?

Line 288- please expand abbreviation FFPE

line 291- amino spelt wrong

Author Response

Thank you very much for the kind review and for the corrections. I have edited the the paper and uploaded an updated copy.

Thank you again,

David

Reviewer 2 Report

undoubtedly this review is complete and thorough, presented in a systematic way with clarity of structural organization of the work, however, I must note that there is no reference to the description of cases of melanoma in cats which may arise in phthysical eyes                  

Author Response

Thank you very much for the review and for pointing out the omission in the paper. I have included information about the development of melanoma in a phthysical eye (line 81).

Thank you again,

David

Reviewer 3 Report

Dear authors,

thank you for this interesting review on feline uveal melanomas.

I have some remarks:

line 25: FDMI should be written out the first time in the text - only in the abstract is not enough.

lline 25+26: delete "old" - it seems to me that is not correct?

line 57: al.

line 62: write out COPLOW

line 103+104: combine the sentences otherwise "ultrasonogrphy" is double

line 126: From my point of view, table 1 is exaggerated, the figures are not so exciting that they have to be presented as a table.

section 2.3: free cDNA: has there ever been an attempt to use cDNA from aqueous fluid of the eye for diagnostic purposes?
And, I think this paragraph would fit better with the section on molecular genetic testing

line 144: the following sections are confuse and should be reordered
3. metastastic diseaes`? - and 3.1. introduction? Most of what comes after that, like immunohisto etc is not limited to metastatic processes.

What is completely missing from my point of view is the pathological diagnosis with its difficulties. e.g. bleaching is often necessary to detect mitoses and nuclear atypia, pigmented cells in the sclera can also be melanophages, pigmented cells often sit around vessels, what is the histological cut off between melanosis and tumor? To what extent can the more advanced methods such as immunohistology and molecular genetics help? Perhaps some collegues from you have example pictures from pathology? And they can give you some advice from trimming the eyeglobe in such cases.

Furthermore, there is a mixture of immunohistochemistry and later you come back to proteins in moleculargenetic section.

line 204: PNLE is an antibody to an antibody????

line 214: specify "other species"

line 232 . at the end of the sentence

line 245-268: From my point of view, this part is much too long - especially because the information that there is no statement on the effectiveness of the therapy comes at the very end!

lines 286-320: In this section, it is again the case that the main critical statement comes only at the end of a long paragraph, namely that it is unclear why the BRAF gene is down-regulated but the proteins are up-regulated - this should be processed more critically in a review directly

line 355 literature number is missing from Rushton

At the end it would be good to give a summary recommendation for the diagnostic, therapeutic and prognostic workup - as far as possible from the data which are available.

Author Response

Thank you very much for the review and for the suggested changes and inclusions.

I have re-ordered the article as suggested and hope that it flows in a more logical manner.

I have emailed my colleagues from the pathology department to request images of sections of FDIM eyes but unfortunately have not heard back yet (I suspect they are still on holiday). I have added a section 'histologic evaluation' to add some information regarding special stains and malignancy criteria for melanocytes. I'm not sure that this fits fantastically well where it is- I initially tried to include it in the introduction where I discuss the cut off between iris melanosis and FDIM but it did not flow nicely. I'm happy to work on this further and take any further suggestions that you may have.

I have cut down the section on the melanoma vaccine as suggested and added a section in the conclusion to include the recommended workup for cases.

With regards to the section on BRAF mRNA vs gene expression- I've cut some information from the section and have added a statement to the end. I hope this is now acceptable? Would it be better to completely cut out the information about why BRAF expression is of interest?

Thank you again for your time and help.

Kind regards,

David

Round 2

Reviewer 2 Report

The work after carrying out the revisions appears complete and detailed, well written and explanatory about a pathology of considerable importance in the clinical practice of small animals

Author Response

Dear reveiwer,

Thank you for your kind words and for your help with the manuscript.

Kind regards,

David Kayes

Reviewer 3 Report

Dear authors,

most of your changes are fine and improve the manuscript.

However, please reconstruct the sections 2.2 and 3.1 and 3.2 because the belong together! I suggest: Iris biopsy - histology/morphology - immunohistology

You cannot separate the diagnosis itself from the criteria of malignancy! For a pathologist this is "inclusive" and there are confusing repetitions in the way you write it actually! This is obvious again if you write about algorhythm of metastastic diseases but the criteria are not clear - the expression of several markers is diferent between benign and malignant diseases?

I suggest:

  • histomorphology for diagnosis of clear pigmented melanocytic neoplasm
  • immunohistochemical markers in amelanotic unclear cases
  • histomorphological mcriteria of malignancy (bleaching, mitoses, morphology) - Form of cells is normally NOT correlated to dignity in melanomas!!!
  • immunohistchocheical markers of malignancy

there a typis in line 158 "Dimmunostaining" and line 180 pseudopolycoria

I understand that you did not get some material from the pathologists - perhaps they are back from holiday now and can help you somehow with the histopath / immunohisto part!

Author Response

Dear reviewer,

Thank you again for your comments and advice. I have re-structured the paper as suggested to flow from iris biopsy to histologic diagnosis and then to immunohistochemical staining. I have removed a section on the morphologic diagnosis from the introduction to include it in this paragraph. I hope this has improved the article from the pathological diagnosis side.

I have managed to get several histology sections that I have included in the article.

I have corrected the typos outlined (now line 167: pseudopolycoria spelling mistake corrected. Now line 255: Immunostaining spelling mistake corrected)

Kind regards,

David